# Use of Cosmetic Creams and Perception of Natural and Eco-Friendly Products by Women: The Role of Sociodemographic Factors

Marina Leite Mitterer-Daltoé *, Vaniele B. Martins, Cristiane R. B. Parabocz and Mário A. A. da Cunha

Graduate Program in Chemical and Biochemical Technology Processes, Department of Chemistry, Federal Technological University of Paraná (UTFPR), Pato Branco 85503-390, PR, Brazil
* Correspondence: marinadaltoe@utfpr.edu.br; Tel./Fax: +55-46-3220-2667

**Abstract:** The present work seeks to understand the use of cosmetic creams and the perception of natural and eco-friendly products by women and to explore the impacts of their personal characteristics. The study was designed using two approaches: (i) an investigation into the role of personal characteristics on the frequency of use and amount spent on cosmetic creams; and (ii) an understanding of the perception of natural and eco-friendly cosmetic creams by the use of check-all-that-apply (CATA) questions. Results showed that scholarity has a strong influence on the use of cosmetic creams; women with a postgraduate education reported higher frequency of use and spending on cosmetic creams and showed an awareness of natural and eco-friendly cosmetics. The subject of natural and eco-friendly cosmetic creams is not something that most of the women that participated were aware of, and the CATA technique proved to be a valuable tool to discover this.

**Keywords:** CATA; green transition; safety assessment; scholarity; skincare

## 1. Introduction

According to The Brazilian Association of the Personal Hygiene, Perfumery and Cosmetics Industry, in the last 3 years, even under the effects of the COVID-19 pandemic, the sale of personal hygiene, perfumery and cosmetic products remained high, with a growth of 4.7% and revenues of USD 25.343 billion in 2020 in Brazil. Skincare products are among those that recorded the highest increases in sales, with a growth percentage of 13% [1,2].

Regarding natural cosmetics, there are no data as to how much this industry markets in Brazil, but some Brazilian brands in this segment reported a growth of 300%. A survey conducted in 2021 showed that products considered eco-friendly saw USD 406 million in distribution in the United States. The study shows that consumers are increasingly spending more on sustainable beauty products, and regarded as the most important those that use natural ingredients (40.2%), respect the environment (17.6%), and use recyclable packaging (7.9% and 15.8%, respectively) [1].

The cosmetic industry is expanding, and along with this, there is a growing demand for natural and eco-friendly cosmetics [3–9], resulting in greater competitiveness in the market [9]. In this context, the development of natural-based cosmetic products can be strategic for this industrial sector. The development of a novel cosmetic product depends not only on the perception of its effectiveness, but also on consumers' perception [9–11]. Consumer understanding is important for both cosmetic companies and consumers. It is important to explore potential market opportunities and satisfy consumers' desires.

The check-all-that-apply (CATA) methodology is a question-based survey format that has been used in recent years to obtain rapid cosmetic product perceptions from the consumer [10,12–14]. Consumers are presented with a list of attributes or sentences and asked to indicate which words or phrases appropriately describe their opinion. The terms

might include sensory attributes, as well as hedonic responses, emotional responses, and purchase intentions, as well as potential applications, product positioning, or other terms that the consumer might associate with the object of the study. A study with industry relevance should probably consider using 100 consumers or more [15]. The selection of CATA terms must be based on previous studies, through attribute survey methodology, for example the Grid Method, or through literature reviews and theoretical foundation. CATA questions are recognized as being quick and easy to carry out by consumers.

In a country like Brazil, where cosmetics use is growing, it is essential to understand the consumption and personal characteristics that impact consumer behavior. In addition, as the natural products industry is a branch of the cosmetics industry in general, which is seeing substantial growth in Brazil and globally, it is necessary to understand the perception of consumers regarding these products. In this sense, the present work seeks to understand the use of cosmetic creams and the perception of natural and eco-friendly products by women and explore decisions based on their personal characteristics. The study was designed using two approaches: (i) an investigation of the role of personal characteristics in the frequency of use and the amount spent on cosmetic creams; and (ii) an understanding of the perception of natural and eco-friendly cosmetic creams through the use of CATA questions.

## 2. Material and Methods

### 2.1. Participants

In the present study, a questionnaire was completed by 192 women from the southwestern region of Paraná, in southern Brazil. In Brazil, women are the vast majority of consumers of cosmetic creams [16], and based on previous studies of cosmetics consumers [10,13] only adult women (aged 18 or over) were asked to participate. The minimum required sample size was determined using the following formula: defining 95% confidence level (z score = 1.96), a standard deviation (SD) of 0.5, and a margin of error of ±5%, and with the total population size of the southwest region of Paraná being 614,000, the minimum required sample size is 385. Considering that only women were the focus of the study, the sample size of 192 participants used in this study was considered to be sufficient.

$$Required\ sample\ =\ \left[z^2\ \mathrm{SD}(1\ -\ \mathrm{SD})\right]/e^2/1\ +\ \left[z^2\ \mathrm{SD}(1\ -\ \mathrm{SD})\right]/e^2\ \mathrm{N}\right] \qquad (1)$$

where N is the population size; z is the z-score; e is the margin of error; and SD is the standard of deviation.

The women were recruited via e-mail from a database. Participants were provided with an informed consent form at the beginning of the questionnaire. The Ethics Committee for Human Research approved the study (CAAE number 2,941,837). Participants were asked information about age, scholarity, occupation, skin type, skin color, and skin problems. Table 1 summarizes the major characteristics of the respondents.

**Table 1.** Characteristics of survey respondents (n = 192).

| Characteristic | | Percentage (%) |
|---|---|---|
| Age (years) | 18–30 | 39.3 |
| | 31–45 | 36.6 |
| | More than 45 | 24 |
| | Secondary (high school) | 18.6 |
| Scholarity (education level) | Tertiary (post high school) | 26.8 |
| | Postgraduate | 54.6 |

**Table 1.** *Cont.*

| Characteristic | | Percentage (%) |
|---|---|---|
| Occupation | Student | 24 |
| | Healthcare professional | 9.3 |
| | Industry | 6 |
| | Commerce | 11.5 |
| | Education | 30.1 |
| | Service provider | 19.1 |
| Skin type | Combined skin | 44.8 |
| | Normal | 25.1 |
| | Dry | 12 |
| | Oily | 18 |
| Skin color | Type I | 14.8 |
| | Type II | 35.5 |
| | Type III | 30.6 |
| | Type IV | 15.8 |
| | Type V | 3.3 |
| | Type VI | 0 |
| Skin problem | Acne | 30.6 |
| | Melasma | 19.7 |
| | Dermatitis | 4.9 |
| | Psoriasis | 1.1 |
| | Eczema | 1.1 |
| | Vitiligo | 0.5 |
| | No problems | 42.1 |

Skin color: Type I (very white) to Type VI (very dark).

## 2.2. Frequency of Cream Usage, Amount Spent, and Check-All-That-Apply Questions

"How frequently do you use cosmetic creams?" was asked of the participants and responses were measured on an 8-point scale: (8) more than once a day, (7) daily, (6) several times a week, (5) weekly, (4) several times a month, (3) monthly, (2) less than monthly, (1) never.

"How much do you spend on cosmetic creams monthly" was asked of the participants and the following options were given: less than USD 10.9; USD 10.9–21.7; and more than USD 21.7.

A CATA question was presented to the participants to assess their perception of natural and eco-friendly cosmetic creams. For each statement, an opposite sentence was presented to obtain reliable results. Participants who marked the statement and its opposite were eliminated. The sentences were presented in a randomized way [17]. The elaboration of sentences was based on a literature review and theoretical foundation. Questions about natural appeal followed previous results [9,18]. In both studies, the subject of natural additives was explored. Ref. [18], using a word association technique, showed a clear potential for the use of natural ingredients, judging by consumer perception that the words were primarily related cognitively to positive aspects of health and food safety. Ref. [9] explored the opinions of potential consumers of cosmetic products containing plant-derived ingredients. The results indicated that the participants see cosmetics with natural ingredients added as being gentle to skin and safe to use.

The sentences about safety and parabens were based on [19–22]. These articles provoke discussion about the use of parabens, which are considered the most widely used preservative by the cosmetic industry, but at the same time are reported as a possible toxicological agent.

The sentences about eco-friendly cosmetic creams were based on [3,7]. Concern about the environment has been increasing and the industrial sector needs to better understand consumers' opinions in order to go through the green transition. Ref. [3] explored ecologically conscious consumer behavior and found that certain demographic variables help differentiate between different segments of green consumers. Ref. [7] explored consumer

resistance toward purchasing eco-friendly cosmetic products. The results highlighted the negative effect of conventional products and the positive effect of health concerns on the desire to consume eco-friendly cosmetics.

Before being questioned, participants were informed of those groups of cosmetics that could be defined as cosmetic creams: body moisturizers, face creams, and hand and foot creams.

### 2.3. Data Analysis

Multinomial logistic regression analysis was applied to evaluate the effects of the independent variables of age, scholarity, occupation, skin type, skin color, and skin problems on the dependent variables of amount spent and frequency of cream usage [23]. The significant variables of global outcomes were also shown by a box plot [24] through the median values.

Analysis of the CATA data was performed by Cochran's Q test carried out amongst all statements with a 5% significance level ($\alpha = 0.05$) [25]. To better understand and visualize the relation between CATA sentences and demographic traits, correspondence analysis (CA) was applied [26]. The data were analyzed using Statistica 12.7 and XLSTAT by Addinsoft.

## 3. Results

### 3.1. The Role of Personal Characteristics in Frequency of Use and Amount Spent on Cosmetic Creams

Figures 1 and 2 show the frequency of use of cosmetic creams and amount spent by consumers, respectively.

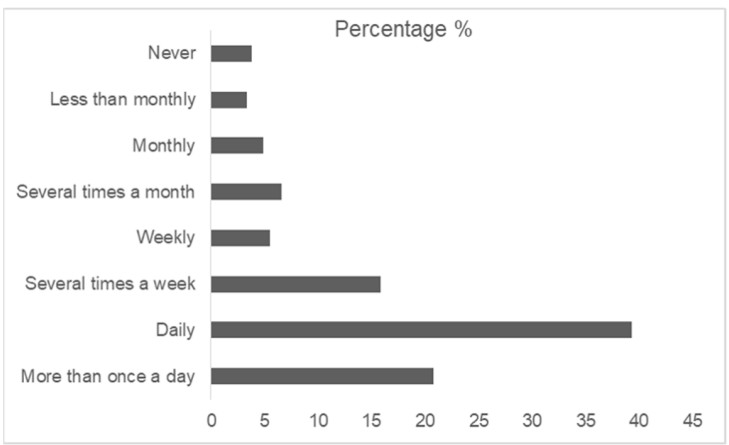

**Figure 1.** Frequency of use of cosmetic creams by women (expressed as percentage). n = 192.

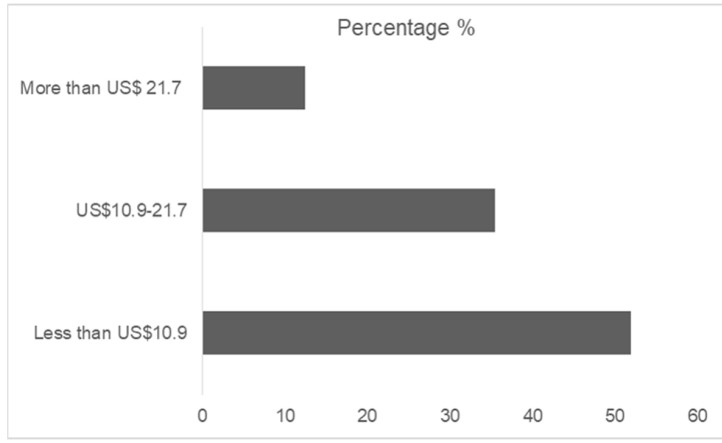

**Figure 2.** Amount spent by women per month (expressed as percentage). n = 192.

The results showed that the frequency of cosmetic cream use is 76.4%, which corresponds, according to the scale (using 100% as the maximum value of the eight-point scale), to a frequency of use between daily and more than once per day. The results from the amount spent related to cosmetic creams per month revealed that 51.9% spend less than USD 10.9, 35.5% spend between USD 10.9 and 21, and 12.5 spend more than USD 21.7.

Aiming to better explore the data, a multinomial logistic regression analysis was performed to relate to the characteristics of survey respondents, the frequency of cosmetic cream use, and the amount spent per month. Multinomial logistic regression is a suitable model in the event that the dependent variable is qualitative. If the logistic coefficient is statistically significant, its interpretation is in terms of how it impacts the dependent variable [27]. Results from Table 2 revealed only the effect of the scholarity ($p \leq 0.05$) on the frequency and amount spent on cosmetic creams. This result suggests an impact of scholarity on the income of respondents and consequently on the frequency and amount spent on cosmetic creams. It is important to note that when applying a simple linear regression, there is not a linear correlation between the frequency of cosmetic cream use and the amount spent, since R square was 0.2933.

**Table 2.** Statistical significance of each variable on frequency of cosmetic cream use and the amount spent per month.

| Effect | Frequency | | Amount Spent | |
|---|---|---|---|---|
| | **Wald** | ***p* Value** | **Wald** | ***p* Value** |
| Age | 3.4073 | 0.3329 | 1.7955 | 0.4074 |
| Scholarity | 7.9857 | 0.0463 | 7.5131 | 0.0476 |
| Occupation | 5.2567 | 0.1539 | 5.0841 | 0.0787 |
| Skin color | 2.6295 | 0.4523 | 1.1609 | 0.5596 |
| Skin problem | 4.7605 | 0.1901 | 1.1609 | 0.5596 |
| Type of skin | 11.4177 | 0.2481 | 5.0054 | 0.5431 |

No significant effects ($p \geq 0.05$) by age, occupation, and all other variables related to skin had an influence on the frequency of cosmetic cream use and the amount spent. These data also revealed important results: women of any age, skin color, skin type, with or without skin problems, and those that carry out any type of work, use and spend money on cosmetic creams to the same extent.

The behavior of the frequency of cosmetic cream use and the amount spent with respect to scholarity can be better visualized by the box plot analyses (Figure 3).

Considering the median values, graphs from Figure 3A showed a predominance of the maximum score (8), indicating the use of cream more than once a day for all scholarities. The post high school group, although showing a median score of 8, presented a lower density in their results, with an intraquartile variance from 6 (several times a week) to 8 (more than once a day), and a minimum recorded frequency of cosmetic cream use of 5 (weekly). This behavior was not registered in other groups. Relative to the amount spent (Figure 3B), the graphs showed a predominance of score 1 for the groups of women with a high school and post high school education, and a predominance of score 2 for postgraduate women. Both graphs suggested a higher frequency of use and spending on cosmetic cream amongst women with a postgraduate education.

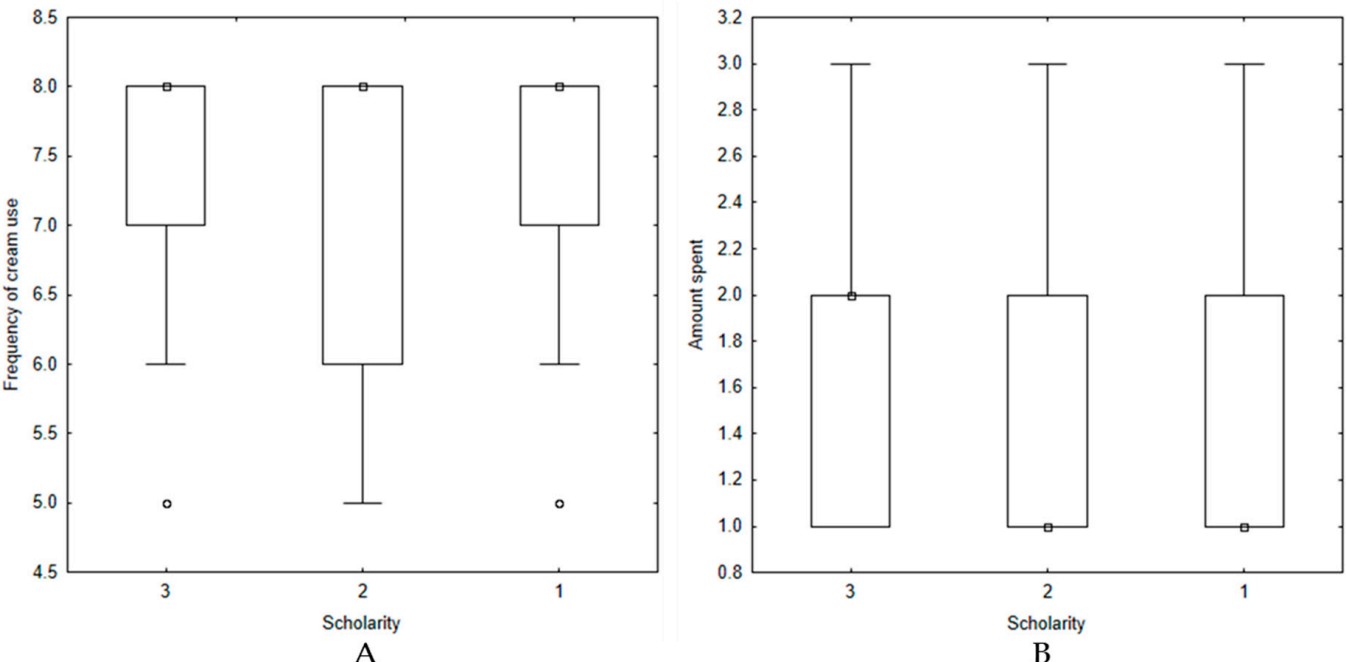

**Figure 3.** (**A**) Box plot of the frequency of cosmetic cream use and amount spent based on level of scholarity. Frequency: (8) more than once a day; (7) daily; (6) several times a week; (5) weekly; (4) several times a month; (3) monthly; (2) less than monthly; (1) never. (**B**) Amount spent: (1) less than USD 10.9; (2) USD 10.9–21.7; (3) more than USD 21.7. Scholarity: (1) secondary (high school); (2) tertiary (post high school); (3) postgraduate. Median: □; 25–75% of the observations: ☐; non-outlier range: ⊥; outlier (extremely high and low values): °.

### 3.2. CATA Questions and Correspondence Analysis to Assess Perception of Natural and Eco-Friendly Cosmetic Creams

Respondents answered the CATA questionnaire that contained sentences designed to explore the perception of natural and eco-friendly cosmetic creams. Table 3 shows the percentage of participants who marked each statement. Significant differences between statements were also obtained by Cochran's Q test.

The analysis of the CATA results indicated that most frequently checked sentences were those related to whether respondents "pay attention to the composition of the creams I use" and "prefer creams that have natural ingredients". Although Q4 was one of the most marked, it should be noted that less than half of the participants checked it. Meanwhile, only 3.3% indicated that they are bothered by the use of creams with natural ingredients, and 6.6% were bothered by the use of 100% natural creams. These results suggest that there is a large group of women that do not prefer cosmetic creams with natural ingredients, but they do not reject them either. All of this can be a consequence of a lack of information or knowledge about natural cosmetic ingredients. The low percentage of checks of the CATA questions in general indicates a lack of critical awareness on this subject. A clear example of this is the analysis of sentences "I do not care how many ingredients the cream has" and "I find it interesting when I see that the cream has few ingredients in its formulation", with 10.9% of marks for both.

This behavior was also evident in the sentences that aim to explore the perception related to eco-friendly cosmetic creams. While just 24.6% of women said they "prefer creams that have an ecological appeal", only 2.7% affirmed that they "do not worry if the production of the cream harms the environment".

**Table 3.** Results of CATA questionnaire. Frequencies (%) for each sentence.

| Code | Question | % | Group |
|---|---|---|---|
| Q1 | I pay attention to the composition of the creams I use. | 43.2 | a |
| Q2 | I am aware of possible adverse health effects caused by the use of parabens and petroleum derivatives. | 24.6 | b |
| Q3 | I do not care how many ingredients the cream has. | 10.9 | bcd |
| Q4 | I prefer creams that have natural ingredients. | 41 | a |
| Q5 | Creams that have natural ingredients bother me. | 3.3 | d |
| Q6 | I find it interesting when I see that a cream has few ingredients in its formulation. | 10.9 | bcd |
| Q7 | It bothers me if creams are 100% natural. | 6.6 | cd |
| Q8 | I prefer creams that are 100% natural. | 14.8 | bcd |
| Q9 | I do not pay attention to the composition of the creams I use. | 17.5 | bc |
| Q10 | I prefer creams that have an ecological appeal. | 24.6 | b |
| Q11 | I do not worry if the production of the cream harms the environment. | 2.7 | d |
| Q12 | I do not know anything about possible of possible adverse health effects caused by the use of parabens and petroleum derivatives. | 11.5 | bcd |

Different letters in the column indicate a significant difference between the number of checks in the check-all-that-apply questions by the Cochran's Q test ($p \leq 0.05$).

Combining the sentences of the CATA questions and the demographic traits by correspondence analysis led to a better understanding of the role of the characteristics of survey respondents on perceptions of natural and eco-friendly cosmetic cream products.

The first two dimensions of the biplots described, from 66.65% to 100%, the correspondence between the characteristics of respondents and the CATA sentences. The lowest variances explained by the first two dimensions were for biplots generated from the skin-related variables (Figures 4–6). Results from other biplots showed greater variances. The analysis showed important results related to sociodemographic traits and the perceptions of natural and eco-friendly cosmetic cream products.

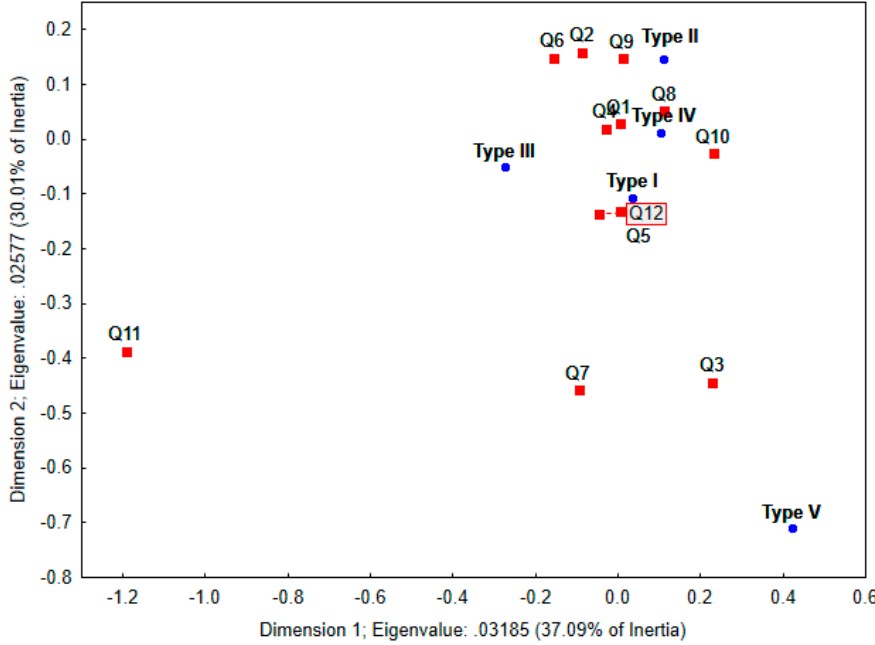

**Figure 4.** Correspondence analysis performed on data from CATA questions and skin color.

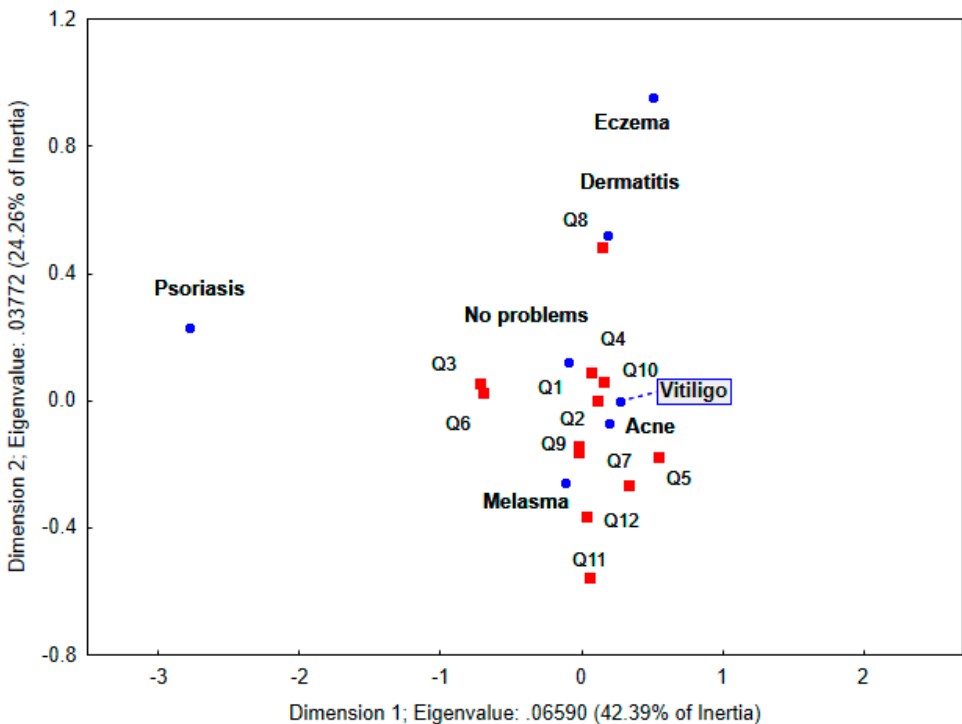

**Figure 5.** Correspondence analysis performed on data from CATA questions and skin problems.

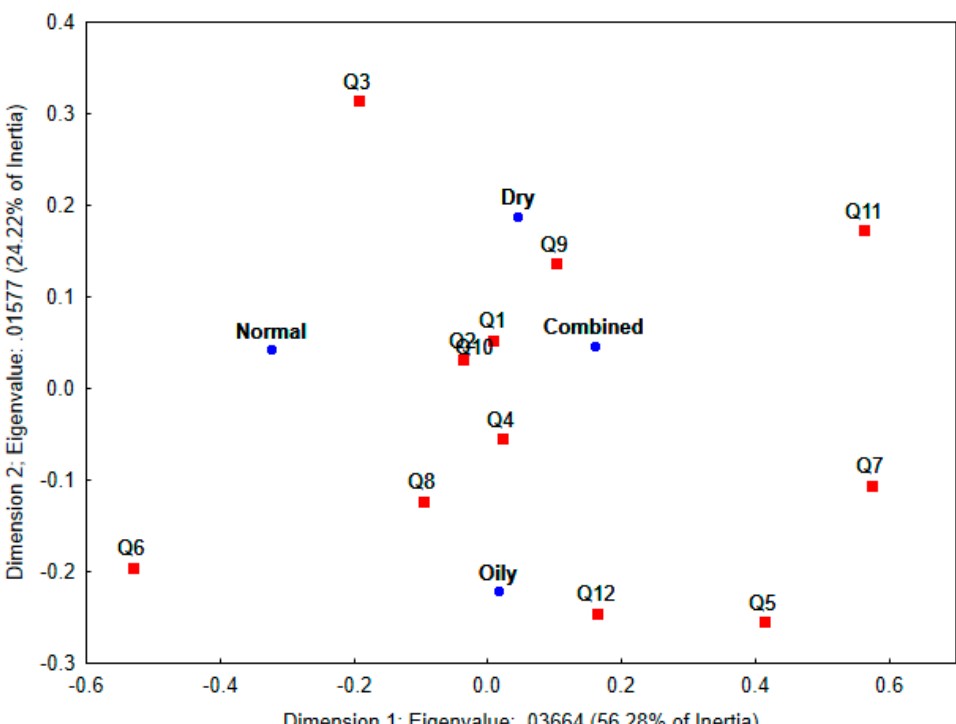

**Figure 6.** Correspondence analysis performed on data from CATA questions and type of skin.

The exploration of the age variable (Figure 7, variance of 100%) showed that older women are more concerned about the use of cosmetic creams that are 100% natural. Conversely, younger ones do not care how many ingredients the cream has aside from affirming that they feel bothered that the creams have natural ingredients. For the amount spent (Figure 8, variance of 100%), women that spent less than USD 10.9 were linked with the statements that reveal that they do not care how many ingredients the cream has; they do

not pay attention to the composition of the creams they use; but it bothers them that creams have natural ingredients. With the group that spent USD 10.9–21, the other statements corresponded better. This group revealed a positive perception about the use of natural and eco-friendly creams.

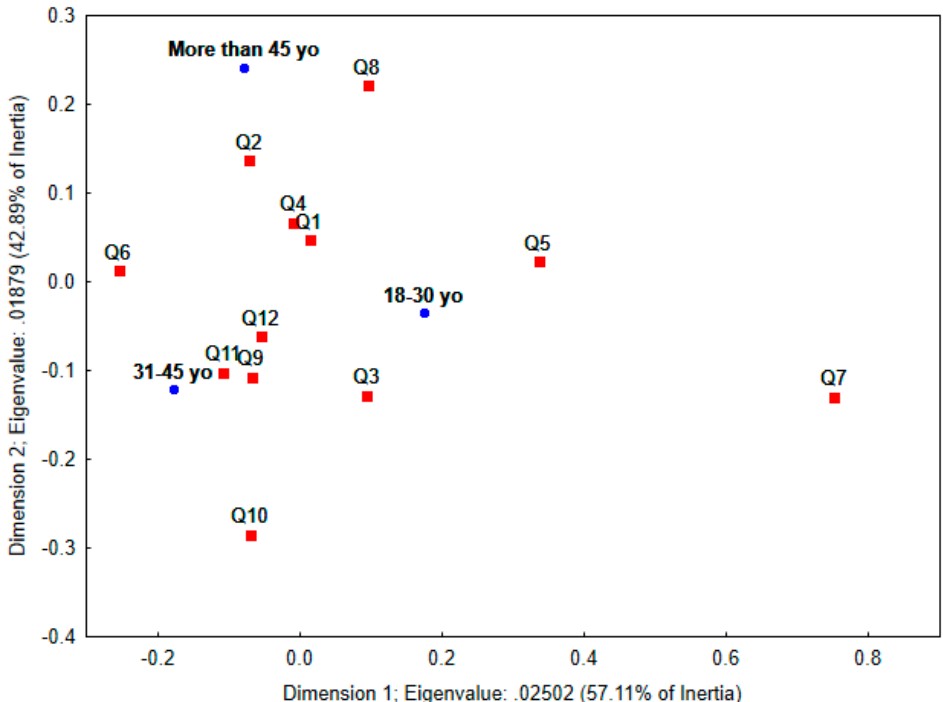

**Figure 7.** Correspondence analysis performed on data from CATA questions and age.

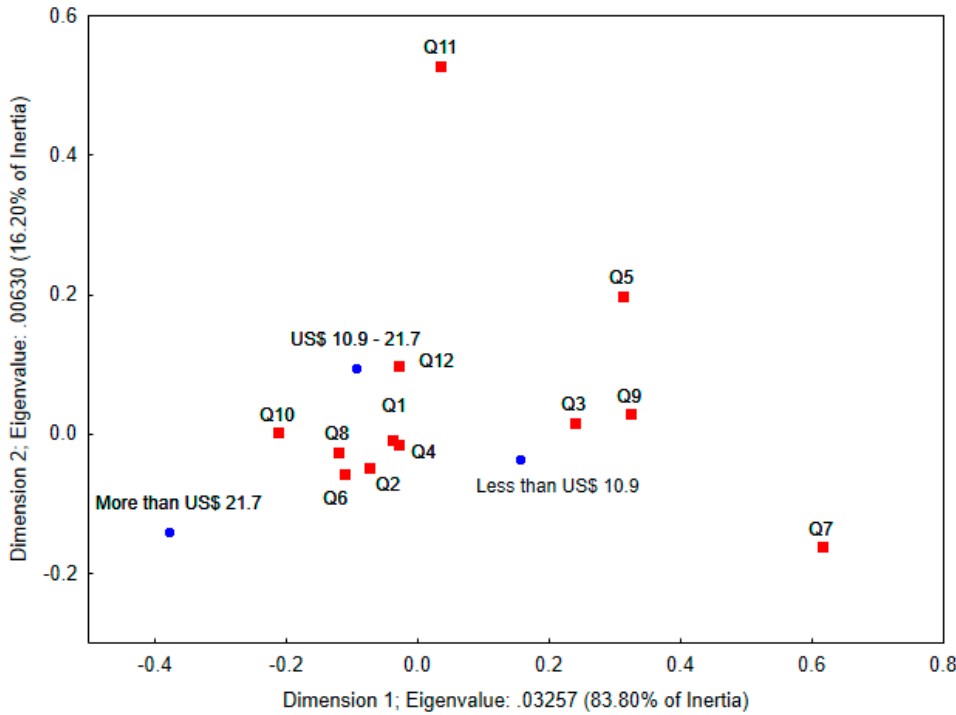

**Figure 8.** Correspondence analysis performed on data from CATA questions and amount spent.

The scholarity data (Figure 9, variance of 100%) showed that only the postgraduate women presented a concern about natural and eco-friendly creams, noting once more a lack of awareness of these issues. Nonetheless, the lack of knowledge about possible adverse

health effects caused by the use of parabens and petroleum derivatives highlights a lack of knowledge by the participants, as even those with more education (postgraduate) were linked to this statement. This issue becomes more evident with the analysis of Figure 10, where not even healthcare professionals knew about the relationship between cosmetic creams and adverse health effects. Still, the results from this biplot are worth mentioning given the importance of natural cosmetics to educated professionals.

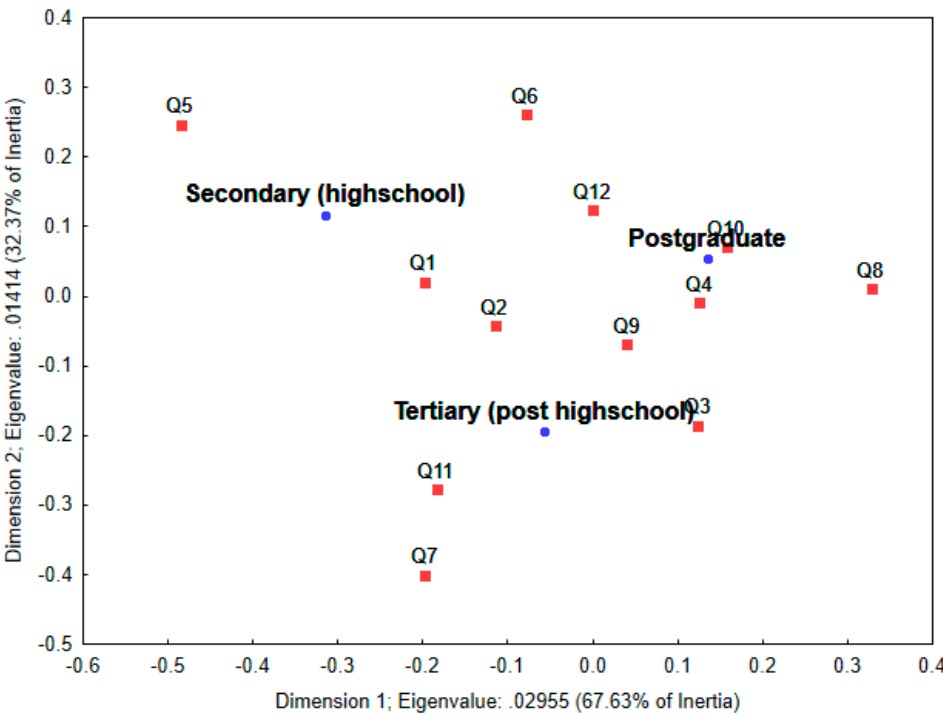

**Figure 9.** Correspondence analysis performed on data from CATA questions and scholarity.

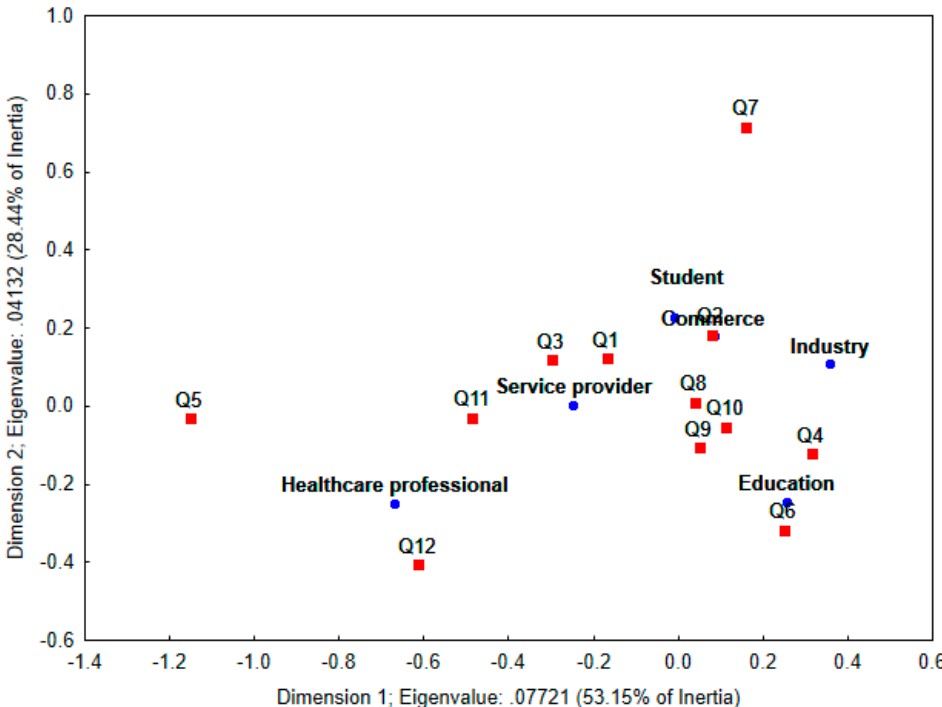

**Figure 10.** Correspondence analysis performed on data from CATA questions and occupation.

Other ingredients, such as isothiazolinones, phenoxyethanol, methylchloroisothiazolinone, methylisothiazolinone, sodium benzoate, benzyl alcohol, and dehydroacetic acid, could have been considered in the study. However, parabens and petroleum derivatives were the objects of this study since, in the past, they were widely used additives and are still used in some cosmetic formulations. In this context, these compounds were only considered as reference ingredients for consumers. Notably, the use of parabens and petroleum derivatives in cosmetic formulations and their adverse effects is still controversial in the scientific community.

Figure 11 shows positive responses indicating that women who more frequently use cosmetic creams are more often interested in creams with natural ingredients and eco-friendly appeal. Women who have the lowest frequency of use, less than monthly, are not concerned about whether or not the production of the cream harms the environment.

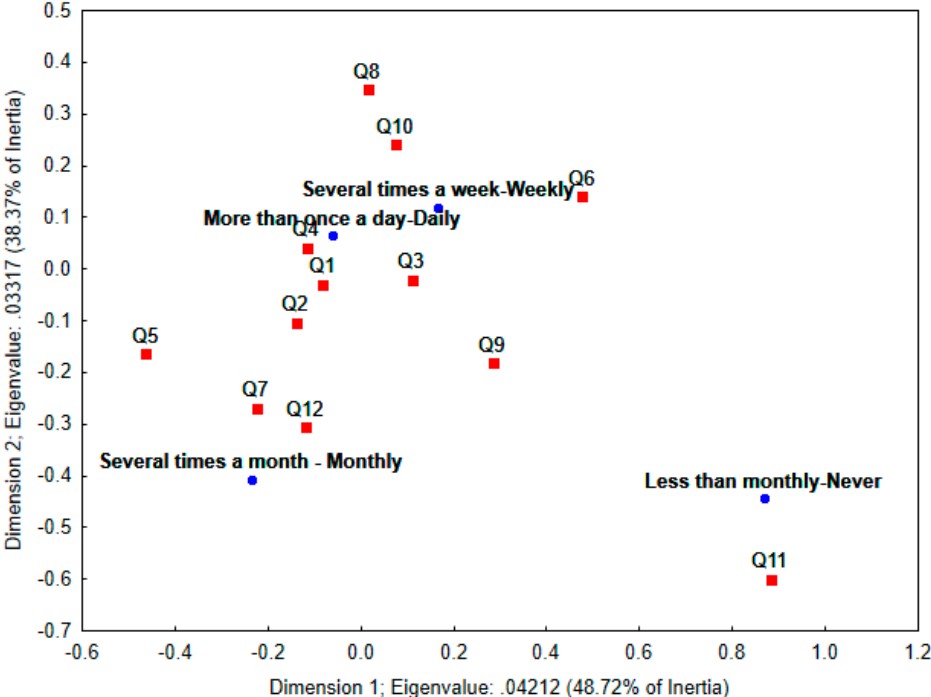

**Figure 11.** Correspondence analysis performed on data from CATA questions and frequency of use of cosmetic creams.

## 4. Discussion

Worldwide, the market for natural and eco-friendly cosmetic products is expanding [28]. In Brazil, this scenario also seems very promising, and for this reason it is essential to understand the use of cosmetics by women and the relationship between the perception of natural and eco-friendly cosmetic products and their personal characteristics. For this research, cosmetic creams were selected as the object of study since they are among the cosmetic products with the highest sales in the country.

Results from multinomial logistic regression showed scholarity as a sociodemographic factor that has an effect on the use of cosmetic creams. Exploring the results by box plots showed a higher frequency of use and spending on cosmetic creams amongst women with a postgraduate education. These results suggested an impact of scholarity on the income of respondents and consequently on the use of cosmetic creams. A previous study investigated which factors affect consumers' purchases of cosmetics by the Theory of Planned Behavior and found that people purchase cosmetics depending on whether they have enough money [11]. In Brazil, the direct relationship between education and income is evident. Accordingly, the Foundation of the Brazilian Institute for Geography and Statistics (IBGE), which compared the average income of Brazilians according to the level

of education, found that professionals with a specialized postgraduate degree, master's or doctoral level, have an income that is 150% to 255% higher than post-high-school-educated professionals. The direct impact of postgraduate studies on Brazilians' salaries influences their purchasing decisions [29].

The fact that other women's personal characteristics did not affect their use of cosmetic creams can be interpreted positively. This demonstrates a large potential market to be explored but one that is affected by affordability. Therefore, this study suggests that a strategy by cosmetic companies should be to try to offer cosmetic creams with more affordable prices, focusing on women with lower incomes.

Until now, the first approach of the study focused on the use of cosmetic creams in general, both synthetic and natural. However, in a population such as Brazil's, the cosmetic industry has scholarity and purchasing power as barriers. When the approach focuses on natural and eco-friendly cosmetics, the issue becomes more important. According to [9], while consumers see plant-derived cosmetics to be gentle to the skin and safe to use, they also see such products as expensive. The authors also found that older respondents with high household incomes were less likely to view plant-based products as expensive. The price issue also becomes important when the subject is eco-friendly cosmetics. The study in [7] showed that environmental concern among consumers indicates that cost has a negative influence and reduces consumers' intention to purchase eco-friendly items.

From this research, the global analysis of CATA demonstrated that most women that participated were not aware of natural and eco-friendly cosmetic creams. The low number of rankings in all sentences highlights the lack of knowledge of these cosmetics. However, it also points out these women as potential consumers of natural cosmetics. Although the majority did not mark the positive sentences related to natural cosmetics, the vast majority also did not mark the sentences that rejected these products. Therefore, the cosmetic companies' strategies should concentrate on making a move to better inform consumers about the composition of cosmetics; the possible adverse health effects caused by the use of parabens and petroleum derivatives; and the sustainable role played by the consumption of eco-friendly products, since consumer knowledge has been found to have positive influence on cosmetic purchase intention [11]. According to these same authors, when consumers think they have higher consumer knowledge, they will have more confidence in their buying decisions, increasing their purchasing awareness.

The joint analysis between CATA and correspondence allowed for an exploration and understanding of the role of women's personal characteristics in the perception of natural and eco-friendly cosmetic creams. This serves as important information for marketers and cosmetic companies to use these findings to promote cosmetic items by giving more information.

The analysis showed important results related to sociodemographic traits and the perceptions of natural and eco-friendly cosmetic cream products. The group of older women (>45 yo), postgraduate, that spent USD 10.9–21 per month on cosmetic creams, and who more frequently use cosmetic creams, revealed a positive perception of the use of natural and eco-friendly creams. Conversely, younger women affirmed the importance of creams that lack natural ingredients. This was an unexpected result, since younger consumers tend to be unconcerned about the potential problems or negative effects of using plant-derived cosmetics [9]. Results from the present research suggest a focused marketing strategy toward young consumers. This will not only promote knowledge in this group but also introduce this knowledge from an early age in this segment of the population, as acquiring a habit requires time and a slow-to-change memory track, which ensures that this learning is retained in the future [30].

Once again, scholarity was an important factor. Only the postgraduate women demonstrated a concern with natural and eco-friendly creams. This highlighted once again the lack of awareness of these issues and the need to provide more information, primarily through marketing campaigns. Despite the large amount of information that is easily available nowadays, consumers do not take the initiative to find it and form their own criteria [31].

Cosmetic products' marketing success depends on an effective link between consumers and the product [10].

The absence of an awareness of the possible adverse health effects caused by the use of parabens and petroleum derivatives emerged as a marker for the lack of knowledge of the advantages of the use of natural cosmetics. Even the postgraduate women and those who were healthcare professionals showed a lack of knowledge on the subject. The safety of cosmetic products is a core topic of cosmetic science and is among the most studied topics in the area of cosmetic technology [21,22]. The field of cosmetic preservatives is an important one and presents challenges to the cosmetics industry and the safety regulatory and dermatology communities [19,20]. Parabens are the preservatives most used in cosmetics and serve two functions: the prevention of product deterioration by microorganisms action, and the prevention of the growth of pathogenic microorganisms [20]. Controversy about parabens by scientists has been reported, and some adverse health outcomes of parabens have been discussed [22]. More importance has been placed on the absorption of parabens, their retention in the human body in their intact form, their toxicological characteristics, and their estrogenic potential [21]. In this scenario, natural preservatives have been gaining great attention in several studies in recent years, as an important healthy alternative to replace synthetic preservatives that have been identified as toxicological agents [4,5,9]; however, it is noteworthy that their use in cosmetics is a challenge [22].

Finally, besides the answers that guided this research, this study provided important insight into the use of the CATA questions in the evaluation of the perception of cosmetic products. The CATA technique proved to be a useful tool to understand the knowledge of a specific topic, but also indirectly showed the lack of it. The absence of markings in both sentences (statement and its opposite) revealed the lack of awareness of the topic. Even so, the joint analysis of CATA questions and correspondence allowed for an understanding of the relationship between knowledge or the lack thereof and the sociodemographic characteristics of the consumers.

## 5. Conclusions

The results showed that the sociodemographic factor of scholarity had an effect on the use of cosmetic creams. Women with a postgraduate education presented a higher frequency of use and spending on cosmetic creams. Scholarity was also an important factor corresponding to the CATA sentences. Again, only the postgraduate women presented a concern about natural and eco-friendly cosmetic creams, although they stated a lack of knowledge about the possible adverse health effects caused by synthetic ingredients in cosmetic creams. The results of this research demonstrated that most women were not aware of natural and eco-friendly cosmetic creams, and the CATA technique proved to be a useful tool to discover this.

Worldwide, the market for natural and eco-friendly cosmetic products is expanding. Understanding consumer behavior and perception is essential for cosmetic companies to create marketing strategies to attract consumers and better explore the green transition and sustainable consumption. These results contribute to the advancement of consumer science, cosmetic science, and companies in the hygiene and beauty sectors.

Future studies exploring which factors influence consumers to purchase cosmetics, and which factors influence the intent to purchase natural cosmetic products, are necessary. Additionally, another important future action would be to investigate the reasons for consumer resistance toward the purchase of eco-friendly cosmetic products.

**Author Contributions:** Conceptualization, M.L.M.-D. and V.B.M.; resources M.L.M.-D., V.B.M. and C.R.B.P.; writing—original draft preparation, M.L.M.-D. and V.B.M.; writing—review and editing M.L.M.-D., V.B.M., C.R.B.P. and M.A.A.d.C.; supervision, M.L.M.-D. and M.A.A.d.C.; project administration, M.L.M.-D., V.B.M. and C.R.B.P. All authors have read and agreed to the published version of the manuscript.

**Funding:** This study was financed in part by the Coordenação de Aperfeiçoamento de Pessoal de Nível Superior—Brasil (CAPES)—Finance Code 001.

**Institutional Review Board Statement:** The Ethics Committee for Human Research approved the study (CAAE number 2941837).

**Informed Consent Statement:** Not applicable.

**Data Availability Statement:** The data presented in this study are available on request from the corresponding author. The data are not publicly available due to ethics committee requirements.

**Acknowledgments:** The authors thank UTFPR-Pato Branco and CAPES and acknowledge the technical support provided. Thanks also to Robert Lee for his assistance with the English manuscript.

**Conflicts of Interest:** The authors declare no conflict of interest.

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
