# Peer review of "Use of Cosmetic Creams and Perception of Natural and Eco-Friendly Products by Women: The Role of Sociodemographic Factors"

_cosmetics, doi:10.3390/cosmetics10030078_

Round 1

Reviewer 1 Report

It is an interesting theoretical research with good scientific soundness regarding the «hot» issue of natural-eco-friendly cosmetic products. However, there are enough important things need correction and further clarification.

Introduction:

Line 70:  Although the focus of study is the women, I believe that it is an error of not using men, for your survey. Please justify it better at the manuscript.

Skin type:  There is no necessity and there are  a lot of mistakes describing the personal characteristics of each type. I prefer to write  again only the simple types (Normal, Dry, Oily, Combined skin) and minimize the table.

Skin color: You describe the common Fitzpatrick classification and it is rather boring and useless. Write only the types without explanation ranging from extremely fair (Type I) to very dark (type VI).

2.2: Line 79-85. You should present the numeric scale in a small Table.

Figure 2: It has to be written per/month.

Figure 4: There are a lot of misleading number, letters etc.

Line: 229: I disagree with the reference of parabens and petroleum for causing adverse events. You have investigated and referred these with way of only these synthetic ingredients are potentially irritant and dangerous, which is a big mistake, according to my opinion.                Other chemicals and derivatives (preservatives, perfumes, PEG, dioxanes, Formaldehyde donors etc) are more suspicious and dangerous. You need to justify the exclusive choice of the above components.

Discussion:  Line 253-259 They are repeated in the introduction!

Line 326-342: I think they should not be written  and deviate from the scientific point of view. Either erase the text or justify adequately.

Author Response

Author Responses to the Reviewers’ Comments

Dear Editor,

Thank you for your comments and suggestions on the structure of our manuscript. We have revised the manuscript according to the reviewers’ recommendations, and responses to reviewers' comments are indicated point-by-point below. 

All changes made to the manuscript have been outlined in the text in red.

REVIEWER’S COMMENTS:

We appreciate the careful assessment and time spent on this review and inform you that all suggested corrections and adjustments have been made. We believe that the manuscript is now suitable for publication after the improvements, and we look forward to a positive response.

Reviewer # 1

It is an interesting theoretical research with good scientific soundness regarding the «hot» issue of natural-eco-friendly cosmetic products. However, there are enough important things need correction and further clarification.

Introduction:

Line 70:  Although the focus of study is the women, I believe that it is an error of not using men, for your survey. Please justify it better at the manuscript.

We understand your point of view and believe that exploring men's perception of natural-eco-friendly cosmetic products will be the subject of future work.

The team of researchers chose to work only with women because in Brazil they represent the vast majority of consumers of cosmetic creams [1].

In addition, with regard to research on cosmetic consumers, there are several works that investigate the perceptions of women only [2–4].

The authors added a justification in the manuscript. See item 2.1.

Skin type:  There is no necessity and there are a lot of mistakes describing the personal characteristics of each type. I prefer to write again only the simple types (Normal, Dry, Oily, Combined skin) and minimize the table.

Thank you for the suggestions. They were essential for the improvement of the work.

Skin color: You describe the common Fitzpatrick classification and it is rather boring and useless. Write only the types without explanation ranging from extremely fair (Type I) to very dark (type VI).

Thank you for the suggestions. They were essential for the improvement of the work.

2.2: Line 79-85. You should present the numeric scale in a small Table.

Figure 2: It has to be written per/month.

Thanks for your suggestion.

Figure 4: There are a lot of misleading number, letters etc.

Figure 4 has been adjusted for better data reading. The revised Figure can be better visualized by the reader, and its data easily interpreted.

Line: 229: I disagree with the reference of parabens and petroleum for causing adverse events. You have investigated and referred these with way of only these synthetic ingredients are potentially irritant and dangerous, which is a big mistake, according to my opinion. Other chemicals and derivatives (preservatives, perfumes, PEG, dioxanes, Formaldehyde donors etc) are more suspicious and dangerous. You need to justify the exclusive choice of the above components.

We agree with your point of view and we are not advocating that parabens and petrolatums cause negative health effects. On the contrary, we show references stating that this issue is questioned by the scientific community.

What motivated us to use them as examples in the study is that these are the synthetic preservatives most used by the cosmetic industry. Therefore, they are supposed to be easier for consumers to refer to. In addition, many cosmetic packages state that they are free of parabens and petrolatums on their labels.

Therefore, our investigation is not about investigating or defending whether these ingredients cause allergies or not but rather investigating whether consumers have some knowledge about these possibilities. Thus, we can better understand and explore the development of natural cosmetics and their market potential.

Discussion:  Line 253-259 They are repeated in the introduction!

Thanks. The paragraph was deleted.

Line 326-342: I think they should not be written  and deviate from the scientific point of view. Either erase the text or justify adequately.

We thank you for your attention to this issue of parabens and petrolatums and we agree with your point of view, for this reason we highlighted in this paragraph that the scientific community finds the adverse effects of parabens and petrolatums to be controversial and merit discussion.

Reviewer # 2

The subject of the paper is in line with the topic of the journal and relevant to the contents of the article. The paper is original, carries significant scientific knowledge, and the results have a practical application, can be used in the cosmetics industry. The manuscript is technically correct. The introduction outlines the problem and the research objective. The research materials and methods are relevant and well-ordered throughout the paper. The paper clearly outlines the method of collecting research materials. The results are complete, appropriately described and novel.  The tables contain all the necessary information and are appropriately captioned. Units and abbreviations are explicit. The reference materials are well-selected and up to date. The references list contains 30 items and it is a representative source for the discussed topic.

In my opinion, the subject matter of the manuscript is appropriate this journal. In general, the topic is interesting, the approach is suitable for a this journal. 
But I have the following comments:

  1. In my opinion, the research sample is too small. Is it possible for 192 people to be representative of the whole population?

The research was based on references that applied the same technique check-all- that- apply (CATA) to consumers to explore their perception of cosmetics. In these studies, there was a variation from 69 [2] to 100 [3] of participating women.

In the book “Novel techniques in Sensory Characterization and Consumer Profiling” edited by Paula Varela and Gastón Ares (2013) [5], Michael Meyners and John C. Castura, they bring the following discussion about design of Cata questions and consumers:

“As with all experiments, the population of interest should be determined, along with consumer selection criteria. Quotas can be set to obtain a stratified sample where appropriate. Consumers should be recruited and selected for the panel or quota group on a random basis, avoiding convenience samples that are unrepresentative of the target population. Sample sizes used in previous studies are quite variable. Both Parente et al. (2010), who compared evaluations of cosmetics on a line scale against evaluations using CATA, and Ares et al. (2010), who studied milk desserts, involved 50 consumers per experimental group. Dooley et al. (2010) used 80 consumers to evaluate ice cream using CATA. Plaehn (2012) used more than 100 consumers to evaluate citrus-flavored sodas. There has not been much investigation to justify an appropriate sample size, but 50 consumers may be low. A study with business relevance should probably consider using 100 consumers or more”.

  1. Why did only women participate in the study? Men use creams too. Perhaps the title of the paper should indicate that it is only about women?

Thanks for your suggestion. We put women in the title.

We understand your point of view and believe that exploring men's perception of natural-eco-friendly cosmetic products will be the subject of future work.

The team of researchers chose to work only with women because in Brazil they represent the vast majority of consumers of cosmetic creams [1].

In addition, with regard to research on cosmetic consumers, there are several works that investigate the perceptions of women only [2–4].

The authors added a justification in the manuscript. See item 2.1.

  1. I get the impression that the publication does not adheres to the journal’s standards. Perhaps text layout is preserved in accordance with the requirements of the editorial, but the formatting is incorrect a few places. For  example table 2 is too large; figure 1 and 2 are of poor quality, Figure 3 has too small letters. This makes analysis difficult; 

Thanks for your suggestion. The quality of tables and figures has been improved as suggested by the reviewer.

  1. His work looks ready to be published, but the English needs to be improved. The language used is correct although it is recommended to read the text by an English-speaking person. 

Thanks. This document was reviewed and edited for proper English language, grammar, punctuation, spelling, and overall style by a native English speaker.

Reviewer # 3

Dear authors, here are my remarks for improve your article:

The required sample formula: the denominator and the numerator are identical, please correct it:

[z2 âˆ™ SD(1-SD)] / e2 / 1 + [z2 âˆ™SD(1-SD)] / e2 âˆ™ N]

Where,

  • N is the population size
  • z is the z-score
  • e is the margin of error
  •   SD is the standard of deviation

Thank you for the suggestions. They were essential for the improvement of the work.

You get the result of 385 if you use the total population and then conclude that: Considering that only women are the focus of the study, the sample size of 192 participants used in this study was considered to be sufficient.” Is the number of women equal with the number of men in Paraná?  If that so, please give a reference that justify your conclusion.

In Paraná (Brazil), the percentage is 51.12% women and 48.88% men [6], corresponding to 196 women from the sample of 385. The researchers of the present study consider that this difference from 196 to 192 does not compromise the current research results.

       Table 1: arrange it so that there is no confusion. For example, on page 2 at the category age it is the subsection Secondary (high school). My advice is to make it more compact (reduce the space between lines).

        Thanks for your suggestion.

”Different letters in the column indicate a significant difference between the number of checks by the Cochran`s Q test (p ≤ 0.05)” -please detail better

       Thanks for your suggestion.

Figures 4-11 – please redo them in order to be clearer.

Thanks for your suggestion. The quality of Figures has been improved as suggested by the reviewer.

REFERENCES

  1. Infante, V.H.P.; Calixto, L.S.; Maia Campos, M.B.G. Comportamento de Homens e Mulheres Quanto Ao Consumo de Cosméticos e a Importância Na Indicação de Produtos e Adesão Ao Tratamento. Surg. Cosmet. Dermatology 2016, 8, 134–141.
  2. Parente, M.E.; Manzoni, A.N.A.V.; Ares, G. EXTERNAL PREFERENCE MAPPING OF COMMERCIAL ANTIAGING CREAMS BASED ON CONSUMERS ’ RESPONSES TO A CHECK-ALL-THAT-APPLY QUESTION. J. Sens. Stud. 2011, 26, 158–166, doi:10.1111/j.1745-459X.2011.00332.x.
  3. Gámbaro, A.; Roascio, A.; Boinbaser, L.; Parente, E. Influence of Packaging and Product Information on Consumer Perception of Cosmetic Creams—A Case Study. J. Sens. Stud. 2017, 32, 1–9, doi:10.1111/joss.12260.
  4. Yano, Y.; Kato, E.; Ohe, Y.; Blandford, D. Examining the Opinions of Potential Consumers about Plant-Derived Cosmetics: An Approach Combining Word Association, Co-Occurrence Network, and Multivariate Probit Analysis. J. Sens. Stud. 2019, 34, 1–9, doi:10.1111/joss.12484.
  5. Varela, P.; Ares, G. Novel Techniques in Sensory Characterization and Consumer Profiling; Varela, P., Ares, G., Eds.; 1st ed.; CRC Press: Boca Raton, 2014; ISBN 978-1-4665-6629-3.
  6. IBGE Brasil e Paraná. Distribuição Percentual de Homens e Mulheres 2010-2016. Brazil and Paraná. Percent Distribution of Men and Women 2010-2016. Available online: https://www.ibge.gov.br/apps/populacao/projecao/box_generos.html?ag=41.

We believe the manuscript is suitable for publication after the improvements are made. We look forward to a positive response.

Thanking you for your kind.

Marina Leite Mitterer-Daltoé

Universidade Tecnológica Federal do Paraná – Brazil

Reviewer 2 Report

The subject of the paper is in line with the topic of the journal and relevant to the contents of the article. The paper is original, carries significant scientific knowledge, and the results have a practical application, can be used in the cosmetics industry. The manuscript is technically correct. The introduction outlines the problem and the research objective. The research materials and methods are relevant and well-ordered throughout the paper. The paper clearly outlines the method of collecting research materials. The results are complete, appropriately described and novel.  The tables contain all the necessary information and are appropriately captioned. Units and abbreviations are explicit. The reference materials are well-selected and up to date. The references list contains 30 items and it is a representative source for the discussed topic. 

In my opinion, the subject matter of the manuscript is appropriate this journal. In general, the topic is interesting, the approach is suitable for a this journal. 

But I have the following comments:

1. In my opinion, the research sample is too small. Is it possible for 192 people to be representative of the whole population?

2. Why did only women participate in the study? Men use creams too. Perhaps the title of the paper should indicate that it is only about women?

3. I get the impression that the publication does not adheres to the journal’s standards. Perhaps text layout is preserved in accordance with the requirements of the editorial, but the formatting is incorrect a few places. For  example table 2 is too large; figure 1 and 2 are of poor quality, Figure 3 has too small letters. This makes analysis difficult; 

4. His work looks ready to be published, but the English needs to be improved. The language used is correct although it is recommended to read the text by an English-speaking person. 

Author Response

(The authors gave the same response as above.)

Reviewer 3 Report

no comments

Author Response

(The authors gave the same response as above.)

Reviewer 4 Report

Dear authors, here are my remarks for improve your article:

1.     The required sample formula: the denominator and the numerator are identical, please correct it:

[z2 ∙ SD(1-SD)] / e2 / 1 + [z2 ∙SD(1-SD)] / e2 ∙ N]

Where,

·        N is the population size

·        z is the z-score

·        e is the margin of error

·        SD is the standard of deviation

You get the result of 385 if you use the total population and then conclude that: Considering that only women are the focus of the study, the sample size of 192 participants used in this study was considered to be sufficient.” Is the number of women equal with the number of men in Paraná?  If that so, please give a reference that justify your conclusion.

2.     Table 1: arrange it so that there is no confusion. For example, on page 2 at the category age it is the subsection Secondary (high school). My advice is to make it more compact (reduce the space between lines).

3.     ”Different letters in the column indicate a significant difference between the number of checks by the Cochran`s Q test (p ≤ 0.05)” -please detail better

4.     Figures 4-11 – please redo them in order to be clearer.

Author Response

(The authors gave the same response as above.)

Round 2

Reviewer 1 Report

I do not see in the manuscript and can not understand why you have not changed and corrected my  remarks regarding the references for parabens-petrochemicals, as I have already written at my first review. Read my previous review for line 229 and lines 326-342. The rest corrections have been done.

Author Response

Author Responses to the Reviewers’ Comments

Dear Editor,

Thank you for your comments and suggestions on the structure of our manuscript. We have revised the manuscript according to the reviewers’ recommendations, and responses to reviewers' comments are indicated point-by-point below. 

All changes made to the manuscript have been outlined in the text in red.

REVIEWER’S COMMENTS:

We appreciate the careful assessment and time spent on this review and inform you that all suggested corrections and adjustments have been made. We believe that the manuscript is now suitable for publication after the improvements, and we look forward to a positive response.

Reviewer # 1

I do not see in the manuscript and can not understand why you have not changed and corrected my  remarks regarding the references for parabens-petrochemicals, as I have already written at my first review. Read my previous review for line 229 and lines 326-342. The rest corrections have been done.

In this second revision, we inserted the following paragraph in the manuscript:

“Other synthetic preservatives such as isothiazolinones, phenoxyethanol, methylchloroisothiazolinone, methylisothiazolinone, sodium benzoate, benzyl alcohol, and dehydroacetic acid could have been in the studies.

However, the objects of this study were parabens and petrolatum because they are the preservatives used most by the cosmetics industry and therefore are thought to be easier for consumers to refer to. It is worth mentioning that the use of parabens and petrolatums, and their adverse effects is controversial within the scientific community.”

Reviewer 4 Report

Thank you for changes.

Author Response

(The authors gave the same response as above.)

Round 3

Reviewer 1 Report

Although, it is not fully justified and nowdays Parabens have been replaced from most cosmetic products in the market, by many alternative preservatives and petrochemicals are not preservatives (please correct it), I consider that the additional paragraph improves the mistake and I accept the manuscript for publication.

Author Response

06 March 2023

RE: Manuscript ID: cosmetics-2289161

Use of cosmetic creams and perception of natural and eco-friendly products by women: the role of sociodemographic factors

Author Responses to the Reviewers’ Comments

Dear Editor,

Thank you for your comments and suggestions on the structure of our manuscript. We have revised the manuscript according to the reviewers’ suggestions, and responses to reviewers' comments are indicated point-by-point below. 

All changes made to the manuscript have been outlined in the text in red.

REVIEWER’S COMMENTS:

Editor comments,

The authors added in the manuscript the sentence below, but petrolatum is an occlusive agent and not a preservative. It also seems that the use of petrolatum is missing from the discussion of results. Please correct the sentence. " However, the objects of this study were parabens and petrolatum because they are the preservatives used most by the cosmetics industry and therefore are thought to be easier for consumers to refer to. It is worth mentioning that the use of parabens and petrolatums, and their adverse effects is controversial within the scientific community."

The authors are grateful for the editor´s comment and inform that the sentence was better written. We believe that the manuscript can now be accepted for publication in its present form.

Synthetic ingredients, such as isothiazolinones, phenoxyethanol,  methylchloroisothiazolinone, methylisothiazolinone, sodium benzoate, benzyl alcohol, and dehydroacetic acid, could have been considered in the study. However, parabens and petrolatum were the objects of this study since, in the past, they were widely used synthetic additives and are still used in some cosmetic formulations. In this context, these compounds were only considered reference synthetic ingredients for consumers. Notably, the use of parabens and petrolatums in cosmetic formulations and their adverse effects is still controversial in the scientific community.

Reviewer #1

Although, it is not fully justified and nowdays Parabens have been replaced from most cosmetic products in the market, by many alternative preservatives and petrochemicals are not preservatives (please correct it), I consider that the additional paragraph improves the mistake and I accept the manuscript for publication.

The authors are grateful for the reviewer's comment and inform that the sentence was better written.